# A Cryptic Invader of the Genus *Persicaria* (Polygonaceae) in La Palma and Gran Canaria (Spain, Canary Islands)

**Filip Verloove** [1,*], **Rainer Otto** [2], **Steven Janssens** [1,3] **and Sang-Tae Kim** [4,*]

1 Meise Botanic Garden, Nieuwelaan 38, B-1860 Meise, Belgium; steven.janssens@botanicgardenmeise.be
2 Lindenstraße 2, D-96163 Gundelsheim, Germany; Rainer.Herta.Otto@t-online.de
3 Institute of Botany and Microbiology, Department of Biology, Katholic University of Leuven, B-3000 Leuven, Belgium
4 Department of Medical & Biological Sciences, The Catholic University of Korea, Bucheon 14662, Korea
* Correspondence: filip.verloove@botanicgardenmeise.be (F.V.); stkim@catholic.ac.kr (S.-T.K.)

**Abstract:** A cryptic invader of the genus *Persicaria* has recently increased in the damper, northern parts of La Palma and Gran Canaria in the Canary Islands (Spain) and locally behaves as an invasive species. Examination of historical herbarium specimens showed this species to be present in Gran Canaria since the 1960s and the same probably applies to La Palma. Up to now, this species had been assigned to the Old World weed *P. maculosa*. However, morphologically, these plants clearly correspond with *P. hydropiperoides*, a common and widespread weed native to the New World, and indeed morphologically similar to *P. maculosa*. Diagnostic features for these two species, as well as for another similar species (*P. decipiens*, originally described from the Canary Islands), are compared, thoroughly discussed, and copiously illustrated. The current distribution, ecology, and naturalization status of *P. hydropiperoides* in the Canary Islands are also assessed. The variability of *P. hydropiperoides* is discussed, more precisely the taxonomic position of a southern 'race' of it that is sometimes referred to as a distinct species, *P. persicarioides*, and to which the Canarian plants belong. The taxonomic value of the latter appears to be clear, although at a lower level. A new combination, at varietal rank, is proposed and validated. In addition to our morphology-based study, a molecular phylogenetic analysis has been conducted on the nuclear ITS region and the plastid DNA region *trnL-F*.

**Keywords:** Canary Islands; cryptic invasion; molecular phylogeny; new combination; nomenclature; *Persicaria*; taxonomy





## 1. Introduction

The generic limits of *Persicaria* (L.) Mill. were controversial for quite a long time. Its segregation from *Polygonum* L., however, is strongly supported by several morphological and anatomical studies [1–4], as well as by recent molecular studies [5–10]. *Persicaria* was placed in the redefined tribe Persicarieae in subfamily Polygonoideae [9] and, in its current circumscription, contains approximately 150 species that are widely distributed in the temperate and subtropical regions of the world. Based on vegetative characters, Haraldson [2] recognized four sections, namely *Cephalophilon* (Meisner) H. Gross, *Echinocaulon* (Meisner) H. Gross, *Persicaria* (Mill.) H. Gross, and *Tovara* (Adanson) H. Gross, which were all well supported by subsequent molecular phylogenetic analyses [5]. In addition, two further sections, *Amphibia* Tzvelev and *Truelloides* Tzvelev, were accepted by Galasso et al. [6]. Of these, *Persicaria* is the largest section consisting of c. 100 species. It is a complex assemblage of polyploids in which hybridization often occurs, e.g., [11–13]. This and uncertain taxonomic boundaries of individual species account for representatives of the section often being badly understood and/or long overlooked, e.g., [14–19]. Yet, many of the species are weedy or may even act as invasive species that enter various kinds of wetland habitats (valley beds, banks of water catchments, etc.).

In the Canary Islands only two species of *Persicaria* section *Persicaria* are known to occur: *P. decipiens* (R. Br.) K.L. Wilson and *P. maculosa* Gray and both are considered to be 'possibly native' [20]. The latter is known from the islands of Gran Canaria, La Palma and Tenerife [20–22]. Since 2002 one of the authors (R.O.) has observed an increasing, weedy species of *Persicaria* in La Palma that somehow resembles *P. maculosa* but subtly differs in a number of characters: plants are rhizomatous perennials rather than annuals (although flowering in the first year), ciliae of the ochrea are longer and nutlets are slightly smaller. Identical plants were recently also found by the first author (F.V.) in a small area in Gran Canaria (surroundings of the Lugarejos lake in Artenara). A revision of herbarium and literature records moreover demonstrated that this species is probably present since at least the 1960s in La Palma and Gran Canaria.

In this paper, the identity of this cryptic invader is discussed, based on morphological data and DNA sequences. It is compared with *P. maculosa* and *P. decipiens* and all three are copiously illustrated. Moreover, the local distribution, habitat preferences, and naturalization status of the species are discussed.

Cryptic invasions are defined as the invasion of non-native species that goes unnoticed due to misidentification as a native or another invasive species [23]. As a result, such events are difficult to recognize and, despite being likely widespread, often go undetected. Their causes and consequences therefore remain largely unknown. Morais and Reichard [23] argued that cryptic invasions may trigger subsequent rapid range expansions and suggested that cryptic invasions are much more common than currently acknowledged. Jarić et al. [24] recently emphasized the importance of crypticity in biological invasions: they may blur invasion impacts and reduce their predictability, since the impacts are often only detected in retrospect and understood with delay, long after control measures would have been effective. They concluded that considering crypticity in biological invasions would strongly enhance the efficiency of monitoring and management planning. We have previously drawn attention to such 'invaders in disguise', as shown by some examples from southern Europe [25]. The present case study is another fine example.

## 2. Materials and Methods

Herbarium specimens of *Persicaria* were examined from the following herbaria: BR, LPA, ORT and the private herbarium of the second author (herbarium acronyms according to [26]) (see Appendix A). Field work was conducted by the second author in La Palma between 2002 and 2019 and by the first author in Gran Canaria in 2017 and 2018. In order to better assess life form and other characters, seeds and underground parts obtained in the field in La Palma were grown ex situ in the garden of the second author in Germany.

Genomic DNA was isolated with an optimized CTAB protocol [27]. Amplification of the plastid *trnL-F* and the nuclear ribosomal ITS region followed [5].

Raw sequences obtained were assembled with Geneious v11.2 (Biomatters, New Zealand). DNA fragments of the *Persicaria* specimens collected by the authors in Gran Canaria and La Palma (Canary Islands), generated in this study, were then combined with sequences from [5,28,29]. The dataset used for further molecular analyses consists of 45 *Persicaria* accessions. *Fallopia scandens* (L.) Holub was chosen as outgroup taxon. Initial alignment was carried out using MAFFT [30] under an E-INS-i algorithm, a 100 PAM/k = 2 scoring matrix, a gap open penalty of 1.3, and an offset value of 0.123. In a second phase, the automatically aligned dataset was manually fine-tuned in Geneious v11.2. Possible phylogenetic conflicts between plastid and nuclear data matrices were inferred by a partition homogeneity test as implemented in PAUP*4.0b10a [31] as well as through visual inspection by searching for conflicting relationships within each topology according to the presence of hard (strongly supported) and soft (weakly supported) incongruences for each node [32]. We selected the best-fit nucleotide substitution model for each gene marker with jModel Test 2.1.4 [33] following the Akaike information criterion (AIC): jModel Test selected GTR+G as most optimal model for ITS and TVM+I+G as most optimal model for *trnL-F*. Bayesian inference (BI) analyses were carried out with MrBayes v3.1 [34]

on individual data partitions. Each analysis was run 10 million generations, with trees sampled every 2000 generations. Chain convergence and ESS parameters were checked with TRACER v1.4 [35]. Bayesian posterior probability (BPP) values between 0.50 and 0.95 (50% majority-rule consensus tree) are regarded as weakly supported, whereas BPP values above or equal to 0.95 are considered as strongly supported [36].

## 3. Results

### 3.1. Taxonomy

3.1.1. Based on Morphological Data

A thorough investigation proved these plants to belong to the American species *Persicaria hydropiperoides* (Michx.) Small, and more precisely to a southern 'race' formerly separated as *P. persicarioides* (Kunth) Small. The following description is entirely based on plant material from La Palma and Gran Canaria:

*Persicaria hydropiperoides* (Michx.) Small, Fl. S.E. U.S. 378. 1903.
≡ *Polygonum hydropiperoides* Michx., Fl. Bor.-Amer. 1: 239. 1803.
Protologue: Hab. in Pensylvana, Virginia, Carolina (U.S.A.). (Type: P).

Plants perennial or annual (seedlings are flowering and fruiting in the first year). Rhizomes usually present, up to 1 cm diameter, often long creeping and generating colonies, in wet or moist substrates by weaker rhizomes and also by means of thinner epigeous creeping stems. Stems often somewhat swollen above the nodes, green, often completely tinged with red or crimson throughout, smooth and hairless, to 1 (–2.5) m, erect or decumbent at the base to ascending, often rooting at the nodes and creeping, simple or branched above the base or richly so from the base. Leaves: ocrea brownish, cylindric to funnelform, 5–25 mm (excl. bristles), chartaceous, margins truncate and ciliate with bristles (2–)3–5(–7) mm long, surface with (0.5–)1(–1.5) mm long, ± distinctly broad-based, coarse, light brownish colored strigose hairs of variable density, not glandular-punctate; petiole 0.5–1.5 (–2) cm, strigose; blade light to deep dark green, sometimes slightly tinged with purple, with (especially when fully exposed to sun) or without a dark blotch adaxially, broadly lanceolate to linear-lanceolate, 5–15(–20) × 0.3–3.5 cm, base tapered or cuneate to acute, margins antrorsely appressed-pubescent, apex acuminate, faces variable regarding their hairiness, nearly glabrous or appressed-pubescent only along midveins or ± strigose allover or only partially on faces, often obscurely punctate abaxially, sometimes densely so. Inflorescences paniculate, composed of usually several spikes-like racemes, these when fully developed (10–)20–40(–60) × 5–8 (–10) mm, erect to somewhat overhanging when fruiting, narrowly oblong to cylindrical and uninterrupted (except sometimes at base), ±obtuse, moderately stout, loose to densely flowered with most flowers not distinct individually and rachis not or hardly visible. Peduncles (5–)10–20(–30) mm, glabrous or strigose; ocreolae funnelform, oblique, ± 3 mm, basally greenish, distally pink, glabrous or more rarely somewhat finely strigose, overlapping distally, sometimes not overlapping proximally, margins ciliate with a few bristles to 0.2–1.5 mm or often naked; pedicels ascending, 2–3 mm. Flowers 2–7(–10) per ocreate fascicle, perianth variously colored, often basally green or pink and distally white, also ± completely white or pink to crimson or spotted, not glandular-punctate; tepals 5, connate to ± less than half length, obovate, 2.5–3(–4) mm, margins entire, apex obtuse to rounded; stamens mostly 6, more rarely 7 or 8, included; style ca. 1 mm, 2- or 3-parted to near the base or near the middle. Achenes nearly always entirely enclosed in the persistent perianth, brownish black or black, (2–)2.1–2.4(–3) mm long, ± smooth (very minutely granular), shiny, narrowly ovoid to broadly oblong in outline, acuminate, mostly lenticular and both sides flat or one side ± angular domed or rarely trigonous, mixed in the same raceme but the trigonous ones are always much less in number.

*Persicaria hydropiperoides* and *P. persicarioides* are closely similar New World species that have been moved back and forth over time. Both were initially described as distinct species by Michaux [37] and Kunth [38] respectively. *P. persicarioides* was found to be most similar to *P. maculosa* (syn.: *Polygonum persicaria*), with which species it shares the predominantly bifid styles and lenticular achenes. The original description, however, is much more

suggestive of *P. hydropiperoides*, as was shown by Stanford [39]. The most notable difference between both was the achene morphology (all trigonous vs. both lenticular and trigonous in *P. hydropiperoides* and *P. persicarioides* respectively) and, to a lesser extent, inflorescence type (somewhat denser in the latter). Since both were shown to have a tendency to intergrade in the southern range of *P. hydropiperoides*, *P. persicarioides* was reduced to a variety of it by Stanford [40]. Subsequent workers accepted this viewpoint, e.g., [41], kept on warranting species status to *P. persicarioides* [42], or subsumed it under *P. hydropiperoides*, as a mere synonym, e.g., [43–45]. *P. hydropiperoides* is known to be a hypervariable species [46].

Table 1 summarizes the most conspicuous differences between *P. hydropiperoides* and *P. persicarioides*, based on data extracted from, among others, [42,47–49] and our own observations.

**Table 1.** Main differences between *P. hydropiperoides* and *P. persicarioides*, based on data extracted from, among others, [42,47–49] and own observations.

| | *P. hydropiperoides* sensu [49] | *P. persicarioides* |
|---|---|---|
| Racemes | Almost linear or narrowly cylindric, +/− interrupted, lax with single flowers +/− distinct and rachis partially visible. In this respect the inflorescence resembles the European species *Persicaria hydropiper* (L.) Delarbre | Narrowly oblong to cylindrical, +/− uninterrupted (except sometimes at base), moderately stout, loose to densely flowered with most flowers not distinct individually and rachis not or hardly visible. The inflorescence resembles the European species *Persicaria maculosa* in this respect |
| Ocrea bristles | 4–10 mm long | (2–)3–5(–7) mm long (reminiscent of *P. maculosa*) |
| Blade | Without dark blotch | With dark blotch (like *P. maculosa*) or sometimes without such a blotch |
| Stamens | 8 | 6–7(–8) (like *P. maculosa*) |
| Style | Homostylous, all styles three-parted | Heterostylous, styles two or three-parted (like *P. maculosa*) |
| Achenes | All trigonous, 1.5–3 mm long, included or apex exserted | Mostly lenticular and both sides flat or one side ± angular domed, more rarely trigonous (like *P. maculosa*), 2.1–2.4(–3) mm long, nearly always included |

The illustrations of *Polygonum hydropiperoides* and *P. persicarioides* in [47] show the subtle but clear difference in inflorescence shape of both taxa, reminiscent of *Persicaria hydropiper* and *P. maculosa* (*Polygonum persicaria*) respectively. Very instructive is also a comparison of the illustration of *Polygonum hydropiperoides* in [48] with the illustration of *Polygonum hydropiperoides* (incl. var. *persicarioides*) in [43]. In the latter, the features of *P. persicarioides* are reproduced exactly.

Based on literature sources, *Persicaria persicarioides* appears to be a more southern species. Historical claims from the United States, e.g., [47] are probably erroneous since the species is not mentioned in the long list of synonyms provided by [49]. *P. hydropiperoides* s.str., in turn, is cold-tolerant and extends north to Alaska and the southern provinces in Canada [49]. Plants from La Palma, planted without protection in a garden in the Bamberg area in southern Germany, did not survive Central European winter conditions, despite repeated attempts.

The plants found in La Palma and Gran Canaria clearly belong to what was formerly called *P. persicarioides* and in several flora accounts fail to key out properly (see for instance the *Persicaria* account in the Flora of North America) [49], for instance because achenes are not strictly trigonous and upper leaf surfaces often have a dark blotch. It therefore seemed appropriate to refer specifically to this identity. However, a combination under *P. hydropiperoides* at the rank of variety seems to be lacking and is thus proposed here:

*Persicaria hydropiperoides* (Michx.) Small var. *persicarioides* (Kunth) Verloove & R. Otto, comb. nov.

Basionym:

*Polygonum persicarioides* Kunth, Nov. Gen. Sp. (quarto ed.) 2: 179. 1817[1818].

TYPE-Protologue: Mexico (Queretaro), crescit in aquis stagnantibus Regni Mexicani prope Queretaro, alt. 995 hex. Floret Junio, Humboldt & Bonpland s.n. (Holotype: P)

Synonyms:

*Persicaria persicarioides* (Kunth) Small, Fl. S.E. U.S. 378. 1903.

*Polygonum hydropiperoides* Michx. var. *persicarioides* (Kunth) Stanford, Rhodora 28(326): 27. 1926.

Furthermore, *Persicaria hydropiperoides,* and especially its var. *persicarioides*, is very similar to *P. maculosa* in many vegetative and floral details, as well as in habit (especially when young). During our study, it became apparent that these taxa have been confused in La Palma and Gran Canaria in the past. The most important diagnostic differences between both are illustrated in Figure 1 and summarized in Table 2, which also includes *P. decipiens* (Figure 2), the only other representative of the section in the Canary Islands, known from La Gomera, Tenerife, and Gran Canaria [20].

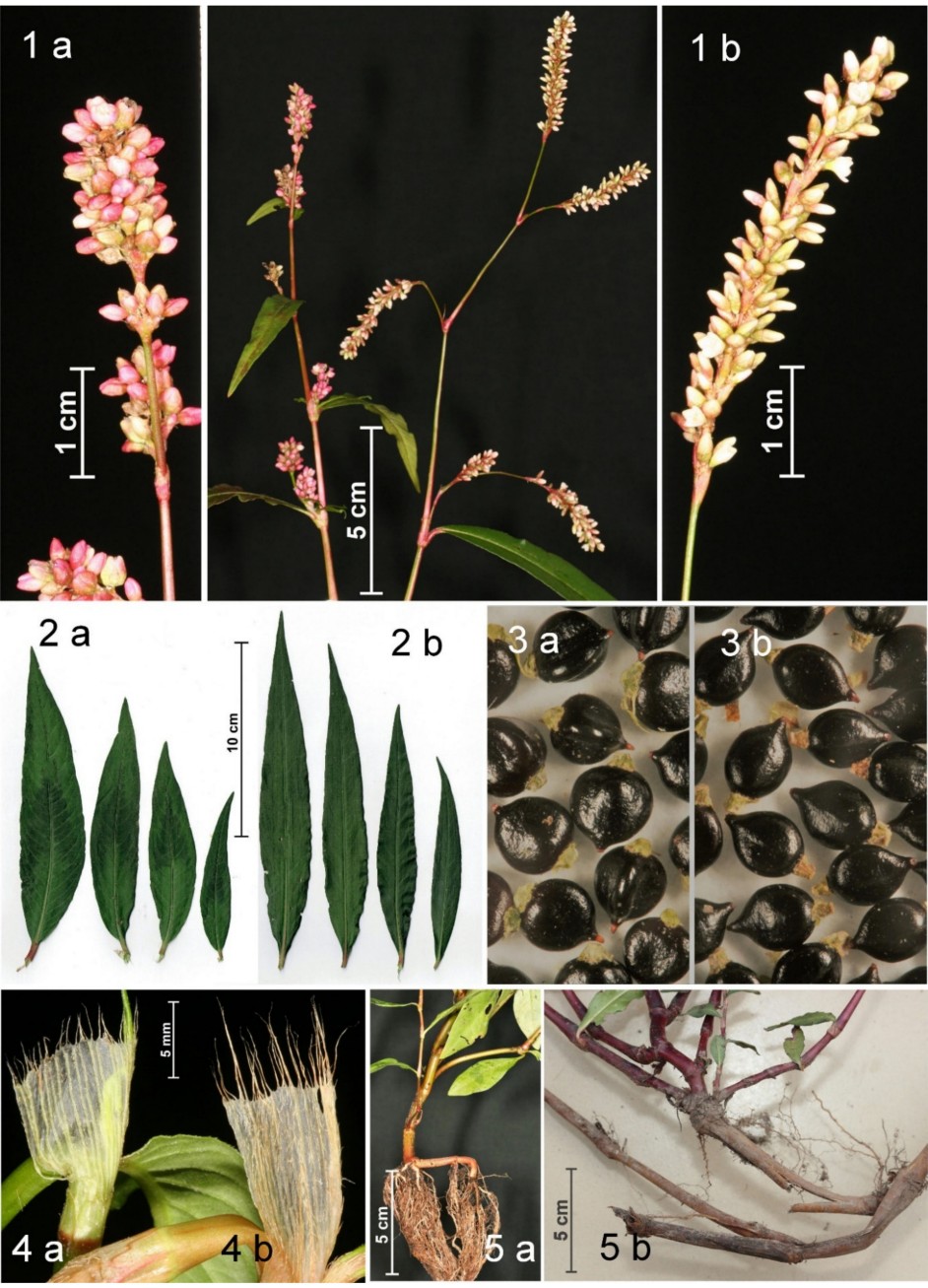

**Figure 1.** Comparison of some diagnostic details of *Persicaria maculata* (**a**) and *P. hydropiperoides* (**b**) (scale identical for both species): (1) fruiting racemes, (2) leaves, (3) achenes, (4) ocrea bristles, (5) subterranean parts. Pictures 1-4 of *P. hydropiperoides* are from plants originating from La Palma and cultivated ex situ in Germany, 5b originates from the plant shown in Figure 4d, pictures of *P. maculata* are from plants collected in Germany (Photographs: R. Otto).

**Table 2.** Differentiating features between the two very variable species *Persicaria maculosa* and *P. hydropiperoides* (var. *persicarioides*), and *P. decipiens*. Data are based on our own observations as well as modified from [47–52]. When using the table, it should be noted that some characteristics can be very differently pronounced in a single specimen of a taxon, are therefore difficult to quantify and may overlap when comparing two species.

| *P. maculosa* | *P. hydropiperoides* var. *persicarioides* | *P. decipiens* |
|---|---|---|
| Annual herb | Annual or perennial herb, seedlings are flowering and fruiting in the first year | Perennial herb |
| Racemes at maturity (5–)10–30(–60) × 7–12 mm, erect, cylindrical and uninterrupted (except sometimes at the very base), obtuse, stout, flowers densely crowded and concealing each other, therefore rachis and each flower not individually distinct | Racemes at maturity (10–)20–40(–60) × 5–8 (–10) mm, erect to pendent, narrowly oblong to cylindrical and ± uninterrupted (except sometimes at base), ± obtuse, moderately stout, loose to densely flowered with most flowers not distinct individually and rachis not or hardly visible | Racemes at maturity 20–60(–90) mm × 6–8 mm, erect, narrowly cylindrical, obtuse, lax and slender, ± interrupted at base, with each flower and the rachis at least partially visible |
| Stem green to brownish, sometimes reddish, to 1 m, erect, ascending or decumbent, simple or mostly ± squarrosely branched, sometimes rooting at nodes | Stem green, often completely tinged with purple red or crimson throughout, to 1 (–2,5) m, erect or decumbent at the base to ascending, often rooting at the nodes and creeping, simple or branched above the base or richly so from the base | Stem green, becoming brown below, to 1 m, erect or basally decumbent, rooting at the lower nodes, simple or sparsely branched, lateral branches strongly upright |
| Rhizomes and stolons absent, root spindle-like | Rhizomes usually present, often thick, long creeping and generating colonies; weaker rhizomes and epigeous creeping stems produced in moist to wet substrate | Rhizomes present, sometimes also stolons |
| Leaf base tapered or cuneate to acute, blade ovate-lanceolate to lanceolate, black blotch adaxially usually present | Leaf base tapered or cuneate to acute, blade broadly-lanceolate to linear-lanceolate, black blotch adaxially present or absent | Leaf base contracted and ± rounded, blade narrowly lanceolate-elliptic, ± long acuminate, black blotch adaxially absent |
| Ocrea glabrous or strigose, often with small, thin, spreading hairs; bristles (0.2–)1.3–2(–3.5) mm long | Ocrea with (0.5–)1(–1.5) mm long, ± distinctly broad-based, coarse, strigose hairs of variable density, rarely glabrous; bristles (2–)3–5(–7) mm long | Ocrea glabrous or ± thinly covered with closely ascending, bristly hairs 1–1.25 mm long; bristles (3–)5–12(–25) mm long |
| Achenes (2–)2.5–2.8(–3.2) mm, mostly lenticular and both sides flat or one side ± angular domed, more rarely trigonous, mixed in the same raceme | Achenes (2–)2.1–2.4(–3) mm, rarely over 2,5 mm long, mostly lenticular and both sides flat or one side ± angular domed, more rarely trigonous, mixed in the same raceme but the trigonous ones always much less in number | Achenes (2–)2.5–3 mm, trigonous (rarely obscurely lenticular). |

In tropical Africa, another very similar species occurs, *Persicaria setosula* (A. Rich.) K.L. Wilson (syn.: *Polygonum setosulum* A. Rich.). It is also weedy [53] and shares many diagnostic features with *P. hydropiperoides*. In fact, initial determination attempts unequivocally led to this species when Old World flora accounts were used, e.g., [50,53,54]. It seems to be differentiated only by the ocrea ciliae that are slightly to markedly longer (2–10 mm long). *P. setosula* also differs from *P. hydropiperoides* var. *hydropiperoides* by its "evenly biconvex-lenticular or . . . obscurely trigonous" nuts [50]. Moreover, in this respect, it is also very similar to *P. hydropiperoides* var. *persicarioides*.

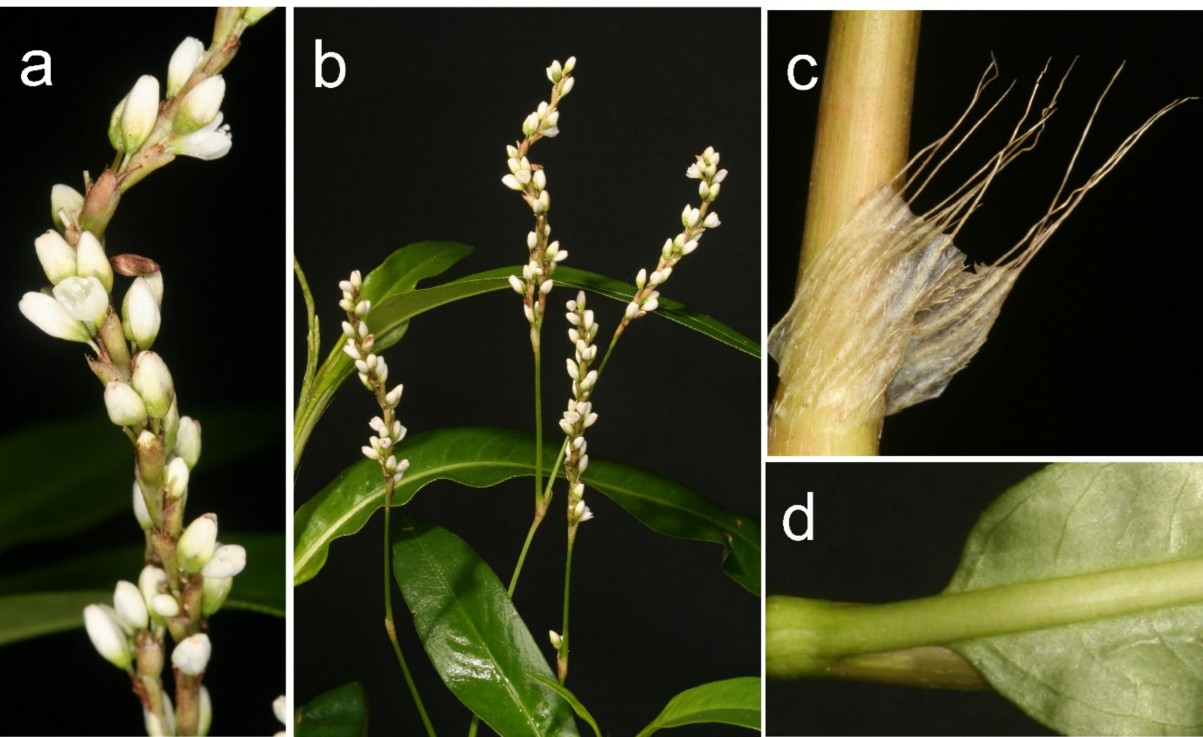

**Figure 2.** Typical characteristics of *Persicaria decipiens* are the slender, loose-flowered (**a**), spiciform racemes 2–10 cm long (**b**). the long ocrea bristles (**c**), and the contracted, almost rounded base of the very short petiolated, sometimes almost sessile, glabrous leaves (**d**). Plants from Sevilla (Spain), cultivated ex situ in Germany, October 2012 (Photograph: R. Otto).

### 3.1.2. Phylogenetic Analyses

The chloroplast *trnL-F* dataset consists of 46 species and 854 analyzed characters of which 116 are variable. The nuclear ITS dataset contains 46 species and 642 analyzed characters (231 variable characters). Visual inspection of the separate *trnL-F* and ITS topologies showed a clear incongruency between the two markers, in which the *P. hydropiperoides* specimens collected in the Canary Islands were positioned in different lineages (Figure 3). The incongruency between both datasets is also confirmed by the ILD test ($p > 0.05$). According to the nuclear topology, the Canary *P. hydropiperoides* are situated as a distinct lineage in a clade with other *P. hydropiperoides*, *P. puritanorum* (Fernald) Soják and *P. opelousana* (Riddell) Small (Bayesian Posterior Probabilities; BPP: 1). The plastid topology indicates a position of the Canary *P. hydropiperoides* accessions as part of a large polytomy including all extant species of *Persicaria* (included in this study) except for *P. amphibia* (BPP: 0.98). However, the Canary *P. hydropiperoides* accessions do not fall together with the other *P. hydropiperoides*, as the latter specimens form a well-supported clade together with *P. opelousana*, *P. hirsuta* (Walter) Small, *P. punctata* (Elliott) Small, and *P. robustior* (Small) E.P. Bicknell (BPP: 0.95).

*Lorem ipsum*

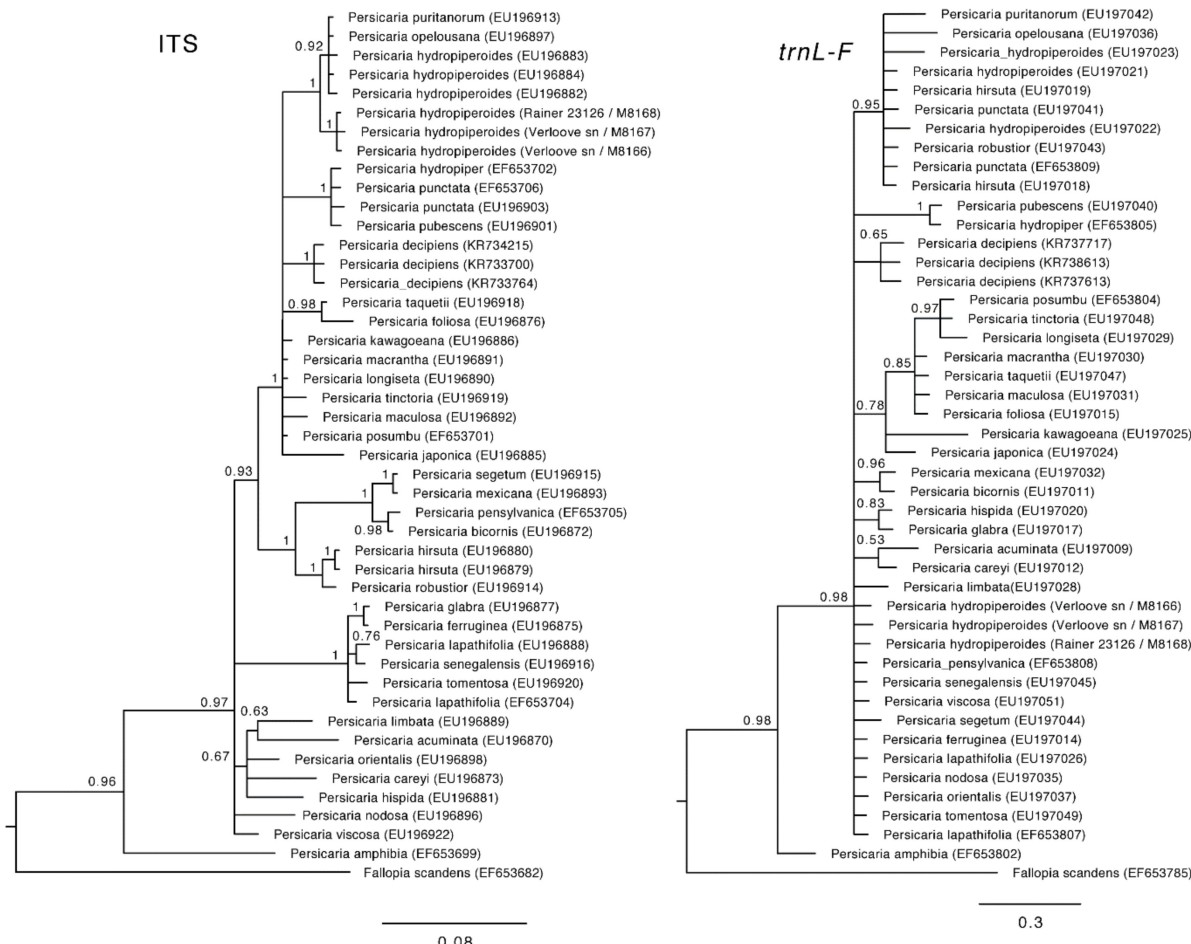

**Figure 3.** *trnL-F* and ITS topologies.

### 3.2. Primary and Secondary Distribution

*Persicaria hydropiperoides* was initially described from the United States [37]. It has a wide distribution throughout America and ranges from Alaska and Canada in the north to Central and South America in the south. It is often considered weedy [55,56] and has been recorded as an alien outside of its native distribution range. According to the Euro+Med PlantBase [57] *P. hydropiperoides* is found in Europe in Transcaucasia (Azerbaijan, Armenia and Georgia). Interestingly, it was introduced long ago and is widely naturalized as a weed in the Azores [58,59]. It was already present there around 1850 and is now considered to be an invasive species [60]. It occurs at altitudes between 100 and 600 m [61] and is found on all islands except Graciosa. It is common on river banks, in marshland and wet pastures [62]. This archipelago in the Atlantic Ocean is part of Macaronesia, to which the Canary Islands also belong.

*P. persicarioides*, in turn, was initially described from Mexico [38] and is further known from Central America (Belize, Costa Rica, Guatemala) and South America (e.g., Argentina, Bolivia; including islands off the coast like Galapagos), e.g., [44,63,64]. To the best of our knowledge, it has never been reported before outside of its natural range, although claims of *P. hydropiperoides* may of course partly refer to it.

In La Palma, the species occurs in the humid northeast of the island, ranging in altitude between a few meters above sea level and 750 m. There are two obvious distribution centers, i.e., the municipalities of Barlovento and San Andrés y Sauces. The largest occurrences

are in the fresh and often foggy, higher locations at the Laguna of Barlovento, one of the areas with the highest precipitation on the island. In Gran Canaria it is well-established in the Lugarejos lake area (Artenara municipality), at an altitude of around 900 m. It is found in Barranco de Lugarejos and Barranco de la Coruña, as well as on the margins of the Lugarejos lake in which both riverlets end.

As shown above, *Persicaria hydropiperoides* can very easily be confused with *P. maculosa*, so it seemed useful to reassess previous claims of the latter in the Canary Islands. It was mentioned for La Palma, Gran Canaria and Tenerife by [20]. For La Palma, the species is reported by [22] from the Barranco del Agua (Los Tilos, 300 m.a.s.l.) and by [21] from the Barranco Herradura (150 m.a.s.l.). These localities all fall within the currently known distribution center of *P. hydropiperoides* on La Palma. We never recorded genuine *P. maculosa* in La Palma in the past decades. Thus, it is possible and even likely that *P. hydropiperoides* has been present on La Palma since the 1960s.

The occurrence of alleged *P. maculosa* in Gran Canaria dates back to [65]: "Barranco del Lugarejo, 900 metros; Ku. 9969. Conocido anteriormente de La Palma." Kunkel pointed out the similarity with *P. salicifolium*: "Similar al *P. salicifolium* Brouss. ex Willd., del cual defiere principalmente por la ócrea que es casi entera (– pronunciadamente ciliata en *P. salicifolium*)". We were able to study Kunkel's collection in LPA and identified it as *P. hydropiperoides*. Moreover, our field study in the Lugarejos area in 2017 and 2018 confirmed the presence of *P. hydropiperoides* while *P. maculosa* was not observed.

From Tenerife, *P. maculosa* was reported from the Barranco Igueste de San Andrés in the northeast of Tenerife by [66] from "Val Igueste", quoted by [67] sub *Persicaria vulgaris*: "ad vallem Igueste insulae Teneriffae". However, the descriptions "Folia iis Salicis viminalis similia" and "Ochreae longè costatae, costis pilosae, longè rigidèque ciliatae" do not refer at all to *P. maculosa*. The narrowly lanceolate leaves and long ochrea bristles would rather suggest *P. decipiens* (syn. *P. salicifolium*) but this species was also treated by Buch l.c., under *P. serrulata* W. et Moq. Perhaps there was, almost two hundred years ago, a confusion with *P. hydropiperoides*. Unfortunately, a search in ORT and LPA for corresponding herbarium specimens was unsuccessful and there might be no extant localities for this species in Tenerife at present.

One can only guess at the origin of the species in the Canary Islands. However, considering the centuries-old very close social and economic relations between the Canary Islands on the one hand and the countries in Central and South America on the other, and the associated exchange of goods, the introduction of the southern variety of *P. hydropiperoides* is not that surprising.

*3.3. Habitat and Ecology*

In its native area, *Persicaria hydropiperoides* is usually found on river and pond margins, in meadows and other, at least temporarily, damp habitats. It is found in similar conditions in the Canary Islands.

In La Palma pure "pioneer" stocks have been observed for example on the dry bare clay masses of a soil depot (Figure 4c,d) and on levelled wasteland next to the large water reservoir in Barlovento. This area was in part used as an unpaved parking and as a storage area for soil excavation. The plants massively grew there on the bare ground with only few and weakly growing accompanying vegetation (e.g., *Cyperus eragrostis* Lam., *Gamochaeta antillana* (Urb.) Anderb, *Laphangium luteoalbum* (L.) Tzvelev, *Mentha pulegium* L., *Polygonum arenastrum* Boreau subsp. *arenastrum*, *Verbena officinalis* L.). The colonization of the area by *P. hydropiperoides* was observed: strong, long underground rhizomes (Figure 1) formed daughter plants (Figure 4c). They richly branched basally and formed compact cushions with decumbent, creeping or obliquely rising stems and branches with short internodes (Figure 4d). The site was later partially rebuilt and lined with hedges. The attempted eradication of the *Persicaria* plants, however, was unsuccessful. The necessary irrigation and nutrient supply to the newly planted hedges immediately led to an extremely lush expulsion of the rhizomes that remained in the soil. Over 1-m-high, upright plants

developed and are a clear indication of the adaptability and variability of this species. As a result of the more intensive use of the site, however, the number of individuals has declined significantly in recent years.

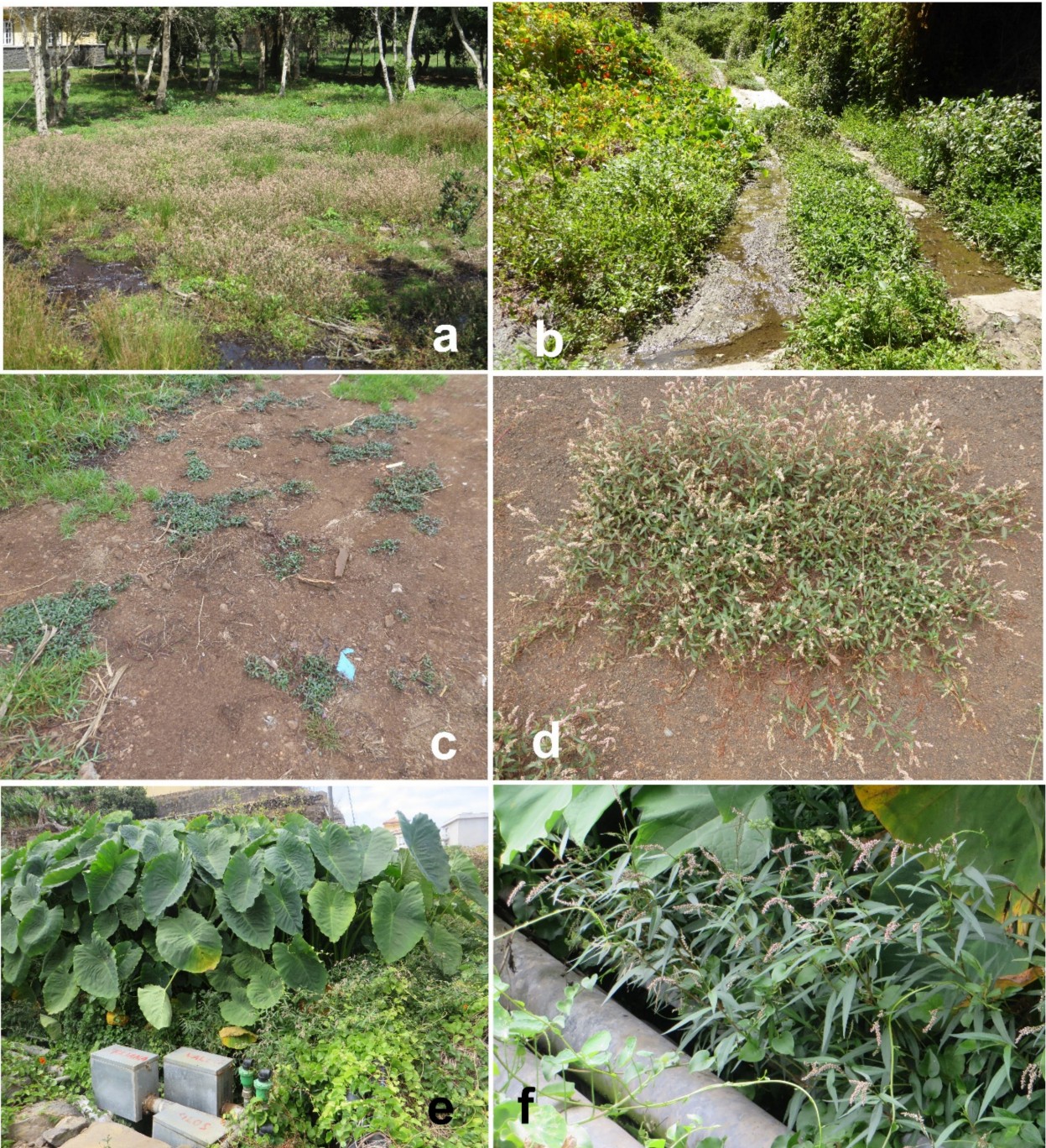

**Figure 4.** *Persicaria hydropiperoides* in La Palma: (**a**) mass occurrence under semiaquatic to moist conditions, Laguna de Barlovento close to the campsite, October 2015; (**b**) typical location of *P. hydropiperoides* in the Barranco San Juan near the mouth, April 2017; (**c**) pioneer situation and mass occurrence in pure stock on a landfill for soil excavation, Laguna de Barlovento, December 2013; (**d**) single plant with several decumbent and semi-erect shoots out of the same rhizome (see Figure 1(5b)) forming a cushion shaped structure ca. 50 cm high and ca. 120 cm diameter, Barlovento, August 2014; (**e**,**f**) occurrence in water channels in the growing area of *Colocasia esculenta*, Los Sauces, August 2014 (Photographs: R. Otto).

Particularly rich in individuals is another stock (approximately 300 square meters) in a somewhat swampy, yearly mown, depression next to the ponds of the leisure facility (Figure 4a). There, the plants are partly standing some cm in water during at least part of the year. Accompanying species are, e.g., *Cyperus eragrostis* Lam., *Hypericum humifusum* L., *Isolepis cernua* (Vahl.) Roem. & Schult., *Juncus effusus* L., *Mentha pulegium* L., *Prunella vulgaris* L. and *Rumex conglomeratus* Murray. During a visit in April 2019, a partial backfilling of the depression with excavation was observed.

As the density of the accompanying ruderal vegetation increases, the growth form changes and the stems are then often more branched in the upper part and reach a length of over two meters (Figure 5b). Such slender and often larger-leaved, green plants are found in places with at least temporarily water supply, e.g., inside or next to water channels and irrigation channels (Figure 4e,f), in road trenches (Figure 5a), next to leaking water pipes and reservoirs, in moist areas with sufficient nutrient supply near to or often below irrigated agricultural cultures, on wet rock walls, and in moist riverbeds of the small and bigger barrancos, as in the Barranco del Agua and Barranco de San Juan near to their mouth. Typical accompanying plants of such wet sites are, e.g., *Ageratina adenophora* (Spreng.) K. King & H. Rob., *A. riparia* (Regel) K. King & H. Rob., *Canna* spec., *Colocasia esculenta* (L.) Schott, *Commelina diffusa* Burm. f., *C. latifolia* Hochst. ex A. Rich., *Cyperus eragrostis* Lam., *C. involucratus* Rottb., *Echinochloa crus-galli* (L.) P. Beauv., *Helosciadium nodiflorum* (L.) W. D. J. Koch, *Polypogon viridis* (Gouan) Breistr., *P.* cf. *fugax* Nees ex Steud., *Tradescantia fluminensis* Vell., and sometimes *Landoltia punctata* (G. Mey.) Les & D.J. Crawford.

Except for *Colocasia* crops in Los Sauces (Figure 4e,f), the species has not been observed as a weed in agricultural crops or in gardens so far.

In Gran Canaria, the species is found in various kinds of at least temporarily damp habitats: gravelly and muddy exposed lake margins, drains and riverlets (Figure 6b), shallow small pools with standing water (Figure 7) in the course of the ravine on the steep hill slope, etc. In the latter kind of habitat, it behaves as a genuine aquatic species. Accompanying species that were observed include, among others, *Ageratina adenophora* (Spreng.) K. King & H. Rob., *Cyperus eragrostis* Lam., *Euphorbia hirsuta* L. (syn.: *E. pubescens* Vahl), *Juncus acutus* L., *Lythrum junceum* Banks & Sol., *Mentha longifolia* (L.) L., and *Nasturtium officinale* R. Br.

### 3.4. Biostatus in the Canary Islands

*Persicaria hydropiperoides* is evidently naturalized in the northeastern part of La Palma where it may be considered a fully invasive species in the sense of [68] (category E): "Fully invasive species, with individuals dispersing, surviving and reproducing at multiple sites across a greater or lesser spectrum of habitats and extent of occurrence". The successful spreading of the plant is enhanced by water transport of the achenes, the transport of mowed plant material of roadsides and by spreading of rhizomes with soil excavation. In Gran Canaria, where it certainly has been present for more than half a century, it is also clearly naturalized. Moreover, in the Lugarejos area it is found in a relatively remote, semi-natural area (Figure 6a). However, as far as we know, all populations seem to be concentrated there in an area of less than 1 km$^2$. As a result, according to [68], it is rather classified as category D2: "Self-sustaining population in the wild, with individuals surviving and reproducing a significant distance from the original point of introduction".

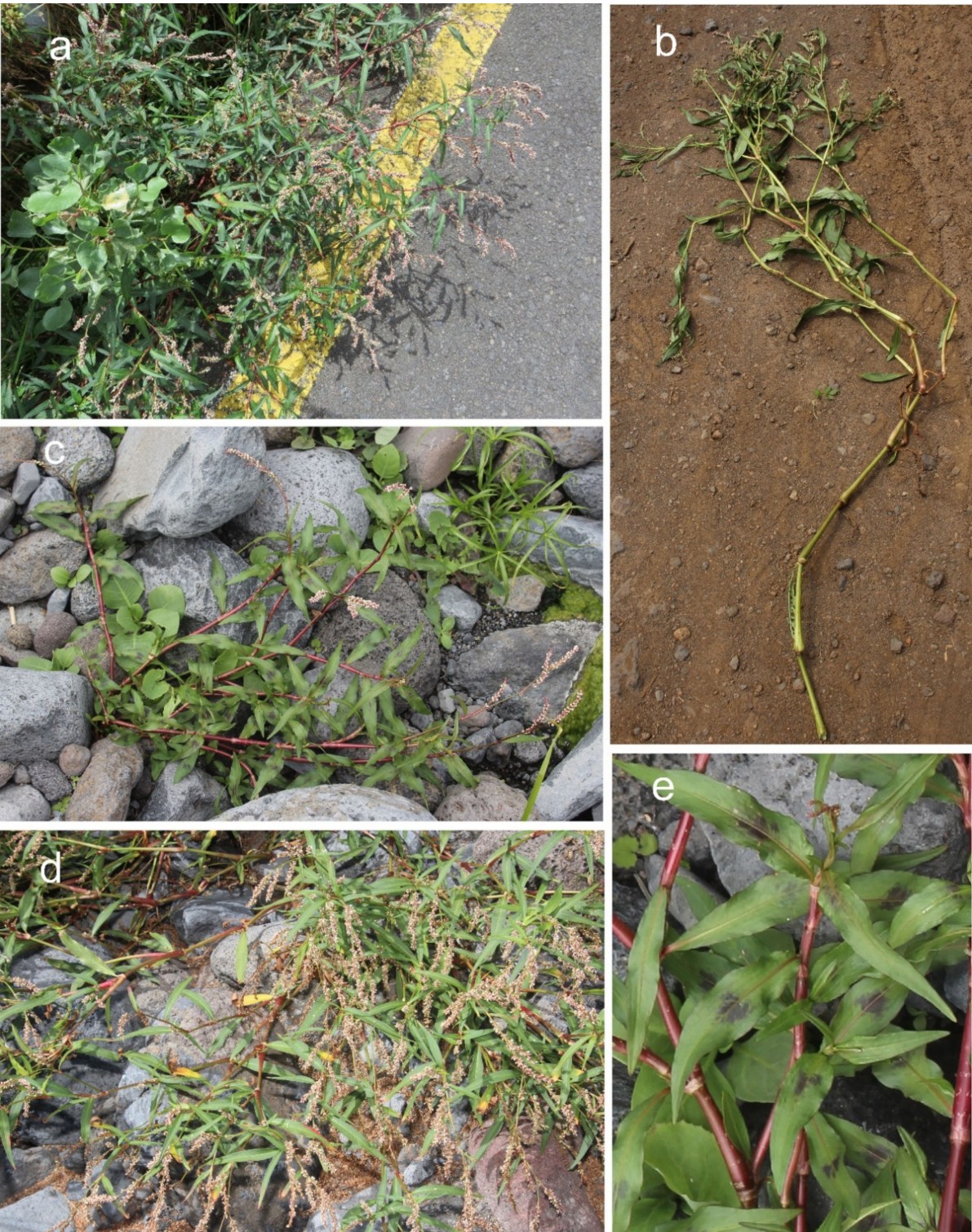

**Figure 5.** *Persicaria hydropiperoides* in La Palma: (**a**) in dense semiaquatic, disturbed vegetation in a moist concreted road trench, Los Sauces, August 2014; (**b**) a stem of such a plant parted from the rhizome neck reaches a length of more than 2.50 m, Los Sauces, September 2013; (**c**) a young specimen in the moist riverbed of the Barranco del Agua in spring, (**d**) the same location as in (**c**) but five months later, *P. hydropiperoides* in the almost dried-out riverbed, procumbent and semi-erect, richly fruiting, October 2013; (**e**) detail of leaf spots (these spots and the intensely purple-red colored stems are typical for sun-exposed plants or parts of plants), May 2013 (Photographs: R. Otto).

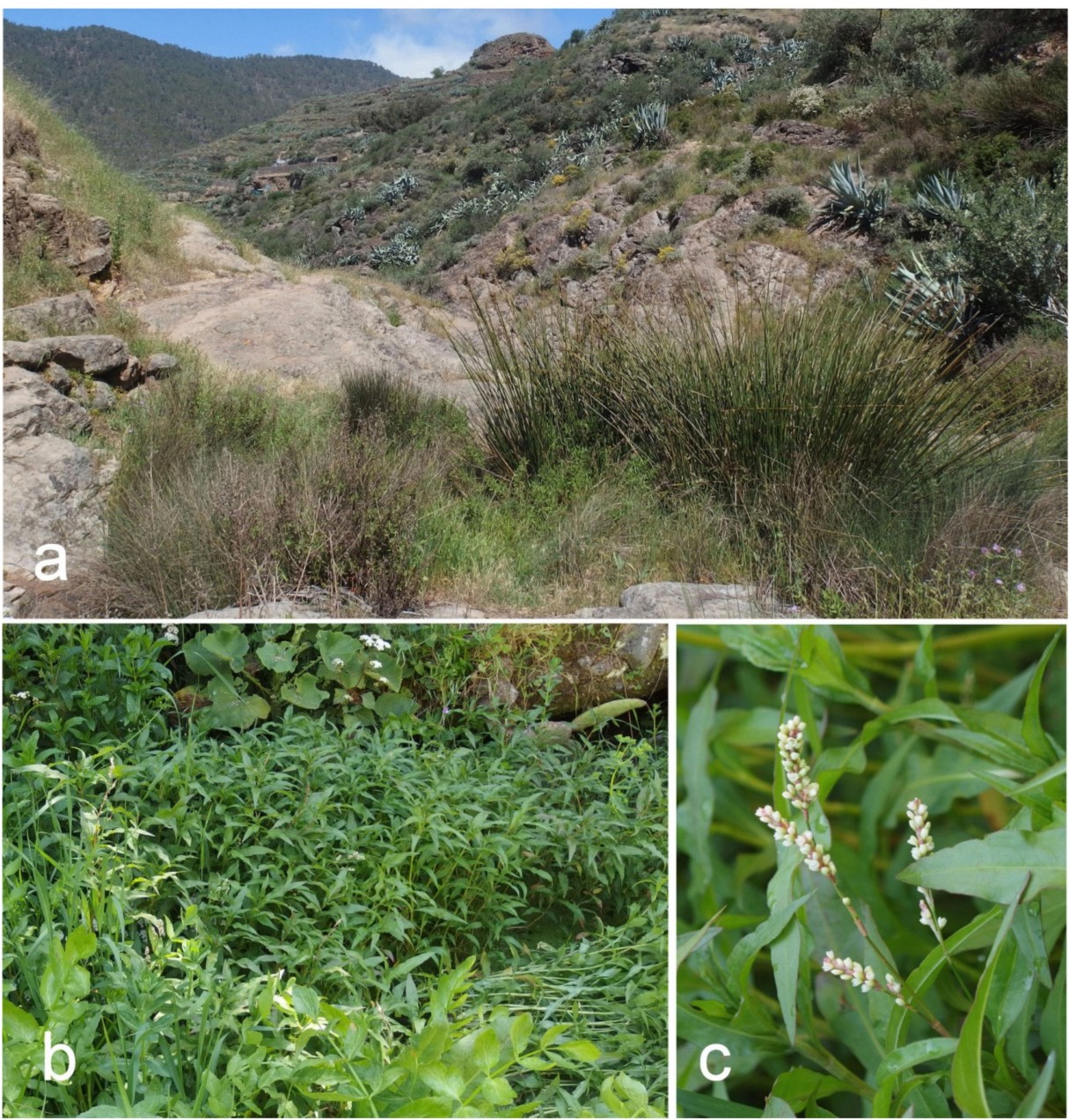

**Figure 6.** *Persicaria hydropiperoides* in Gran Canaria, Barranco de Lugarejos, April 2017: (**a**) general view of the rather remote and natural area in which the species is naturalized, (**b**) the species co-dominates the vegetation in riverlets and drains, (**c**) detail of inflorescence (flowers can range in color from pink to white) (Photographs: F. Verloove).

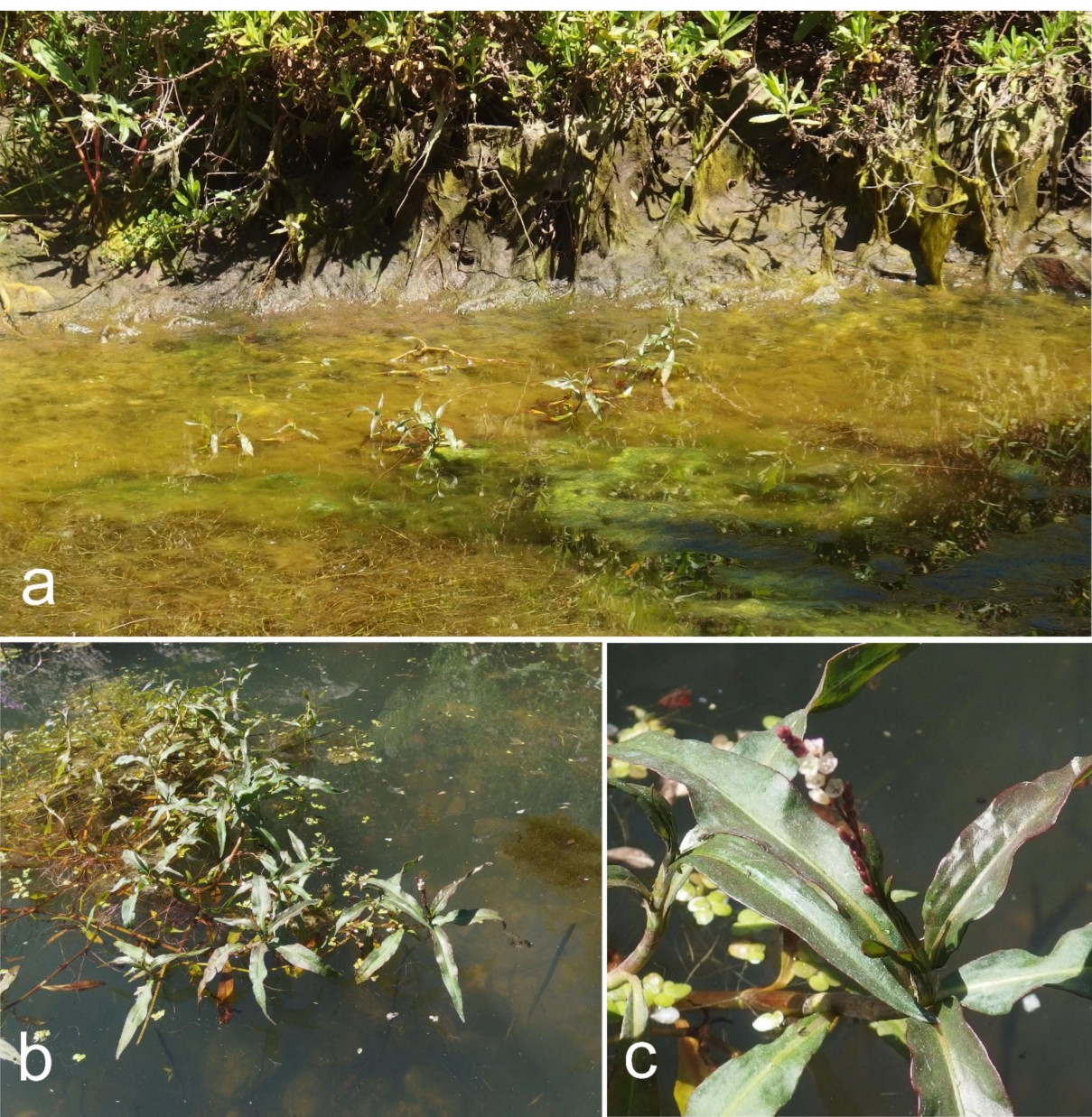

**Figure 7.** *Persicaria hydropiperoides* in Gran Canaria, Barranco de Lugarejos, April 2017: (**a**–**c**) in temporary pools the species behaves as a genuine aquatic species (Photographs: F. Verloove).

## 4. Discussion and Conclusions

Canarian populations of a species of *Persicaria* formerly thought to belong to the putatively native species *P. maculosa* were shown to pertain to the introduced New World weed *P. hydropiperoides* instead. Morphologically, these plants can be ascribed to a southern race that naturally occurs in Central and South America and that is sometimes accepted as a distinct species, *P. persicarioides*.

While by virtue of morphological features the identity of the Canarian plant material appears to be straightforward, there are discrepancies as far as their genetic identity is concerned. The chloroplast *trnL-F* and nuclear ribosomal ITS datasets were incongruent as far as the position of Canarian accessions of *P. hydropiperoides* are concerned. Whereas the nuclear topology places Canary *P. hydropiperoides* in a distinct lineage in a clade that comprises New World accessions of *P. hydropiperoides* (but also of related species such as *P. puritanorum*, *P. mitis*, *P. minor*, and *P. densiflora*), the plastid topology indicates a position

as part of a large polytomy, together with morphologically very diverse species, many of them only remotely related to *P. hydropiperoides*.

This incongruence among, on the one hand our chloroplast and nuclear gene trees and on the other hand our morphological data, is not easily explained. However, Kim and Donoghue [5] already demonstrated that the *P. hydropiperoides* complex, including *P. hydropiperoides* and *P. opelousana* (Riddell) Small, probably originated via allopolyploid speciation. Multiple instances of hybridization and allopolyploidy have occurred in this group and this has contributed to the taxonomic difficulties and to the incongruence of molecular and morphological data. In Kim and Donoghue's phylogenetic analysis of 2008 of the combined chloroplast DNA data set, three accessions of *P. hydropiperoides* did not appear together. One formed a clade with *P. setacea* (Baldwin) Small and *P. hirsuta* (Walter) Small and another with *P. puritanorum* and *P. punctata* (Elliott) Small, whereas the position of the third accession remained unresolved.

Genetically, the Canary accessions of *P. hydropiperoides* appear to be not exactly identical with New World accessions of that species. However, the latter all originate from North America (Connecticut and Florida) and thus belong to the 'typical' (North American) form of that species. As demonstrated above, the Canarian plants are slightly but definitely aberrant because they do not fully correspond morphologically with that northern form but perfectly agree with the southern *P. persicarioides* for which taxon no accessions seem to be available in Gen Bank. This presents all the more reason to resurrect the latter and recognize it as a separate taxon, albeit at a lower taxonomic rank.

**Author Contributions:** Conceptualization, writing—original draft preparation, and fieldwork, F.V. and R.O.; molecular analysis and review and editing S.J. and S.-T.K. All authors have read and agreed to the published version of the manuscript.

**Funding:** Fieldwork by the first author in March and April 2017 in Gran Canaria was granted by COST-Action TD1209 (reference: TD1209-290317-084724).

**Institutional Review Board Statement:** Not applicable.

**Data Availability Statement:** The sequencing data generated during this study were deposited at GenBank but their accessions numbers were not yet available at the time of publication.

**Acknowledgments:** Rolf Wisskirchen (Germany) and Diego Giraldo-Cañas (Colombia) are sincerely thanked for their assistance with the identification. The curators of the herbaria LPA (Águedo Marrero Rodríguez) and ORT (Jorge Alfredo Reyes-Betancort) are acknowledged for assisting the first author during his herbarium revision. Finally, Wim Baert (Meise Botanic Garden) is thanked for his assistance in the lab.

**Conflicts of Interest:** The authors declare no conflict of interest.

### Appendix A. Specimens Examined

From La Palma (Spain, Canary Islands):

San Andrés y Sauces, Los Sauces, Las Lomadas, road trench, 18.08.2002, *R. Otto* 7832 (pers. herb. RO); ibid., LP-1, roadside, road trench 15.08.2003, *R. Otto* 8539 (pers. herb. RO);

Barlovento, next to Laguna de Barlovento, landfill for soil (excavation) and rubble, in the edge area also used as a landfill for green waste, several individuals on bare ground, ca. 735 m, 28°48′37.5″ N 17°48′11.0″ W, 27.09.2007, 02.10.2012, *R. Otto* 10120 (pers. herb. RO);

San Andrés y Sauces, Los Sauces, concrete moat roadside LP-1 opposite Ermita Santa Rita, many individuals, ca. 188 m, 28°48′45.3″ N 17°46′25.7″ W, 02.10.2012, *R. Otto* 19830 (pers. herb. RO); ibid., 05.08.2014, *R. Otto* 21186 (pers. herb. RO, dupl. BR);

San Andrés y Sauces, San Andrés, Barranco del Agua, gravelly riverbed, moist, ca.150 m before mouth, several individuals, ca. 20 m, 28°48′03.1″ N 17°45′37.9″ W, 26.05.2013, *R. Otto* 20265 (pers. herb. RO);

Barlovento, alongside LP-109 approximately 100 m north turn-off of the road to Laguna de Barlovento, landfill for soil (excavation) and rubble, several individuals on bare

ground, 28°49′02.8′′ N 17°48′41.3′′ W, 27.05.2013, *R. Otto* 20292 (pers. herb. RO, dupl. BR); ibid., 06.10.2015, *R. Otto* 21913 (pers. herb. RO);

San Andrés y Sauces, Los Sauces, Calle 1a, under and out of wet rock face grow several individuals, 290 m, 28°48′15.5′′ N 17°46′31.2′′ W, 02.06.2013, *R. Otto* 20354 (pers. herb. RO);

San Andrés y Sauces, San Andrés close to Llano el Pino, alongside LP-104, concreted road trench (often water bearing), ca. 175 m, 28°47′35.1′′ N 17°46′03.1′′ W, 20.09.2013, *R. Otto* 20632 (pers. herb. RO, dupl. BR); ibid., 02.08.2014, *R. Otto* 21169 (pers. herb. RO); ibid., 03.11.2014, *R. Otto* 21320 (pers. herb. RO);

Barlovento, next to Laguna de Barlovento, landfill for soil (excavation) and rubble, mass occurrence on bare ground, ca. 735 m, 28°48′40.8′′ N 17°48′06.9′′ W, 25.09.2013, *R. Otto* 20676 (pers. herb. RO, dupl. BR); ibid., 05.08.2014, *R. Otto* 21198 (pers. herb. RO, dupl. BR);

Barlovento, next to Laguna de Barlovento, unpaved parking, foot of the hedge around the fairground, the hedge was new planted and irrigated from time to time, mass occurrence on bare ground, ca. 730 m, 28°48′42.8′′ N 17°48′10.3′′ W, 25.09.2013, *R. Otto* 20680 (pers. herb. RO); ibid., 03.11.2014, *R. Otto* 21322 (pers. herb. RO);

Barlovento, Laguna de Barlovento, next to recreational facility, damp earthy road trench, ca. 730 m, numerous, 28°48′30.9′′ N 17°48′18.2′′ W, 25.09.2013, *R. Otto* 20705 (pers. herb. RO, dupl. BR);

Barlovento, Laguna de Barlovento, next to the recreational facility, yearly mown wet and marshy depression of some hundreds square meters between parking and camping place, plants stand partially 5–10 cm in the water, mass occurrence, ca. 730 m, 28°48′32.7′′ N 17°48′12.4′′ W, 02.10.2015, *R. Otto* 21891 (pers. herb. RO, dupl. BR); ibid., 01.04.2017, *R. Otto* 22511 (pers. herb. RO, dupl. BR); ibid., 02.04.2018, *R. Otto* 23126 (pers. herb. RO, dupl. BR);

San Andrés y Sauces, near the cemetery of San Andrés, damp riverbed of the Barranco de San Juan ca. 0,5 km before the mouth, overgrown river bed is used as unpaved road, ca. 38 m, ca. 28°47′26.8′′ N 17°45′42.1′′ W, 04.04.2017, *R. Otto* 22547 (pers. herb. RO, dupl. BR).

From Gran Canaria (Spain, Canary Islands):

[Artenara], Bco. [Barranco] del Lugarejo, parte humeda, 900 m.a.s.l., 27.01.1967, *G. Kunkel* 9969 (sub *Polygonum persicaria*; LPA);

Artenara, Lugarejos, Barranco de Lugarejos, 28°2′30.88′′ N, 15°40′13.67′′ W, 891 m.a.s.l, shallow pool close to the lake, 07.04.2017, *F. Verloove* 13699 (BR);

Artenara, Lugarejos, Barranco de Lugarejos, 28°2′21.03′′ N, 15°39′57.81′′ W, 955 m.a.s.l, riverlet, 07.04.2017, *F. Verloove* 13700 (BR);

Artenara, Lugarejos, Barranco de Lugarejos, 28°2′16.84′′ N, 15°39′56.31′′ W, 956 m.a.s.l, riverlet, 07.04.2017, *F. Verloove* 13701 (BR);

Artenara, Las Hoyas, Presa de Lugarejos, exposed pond margin, 28° 2′29.20′′ N, 15°40′32.92′′ W, 905 m.a.s.l, riverlet, 07.04.2017, *F. Verloove* 13705 (BR).

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
