# Peer review of "A Cryptic Invader of the Genus Persicaria (Polygonaceae) in La Palma and Gran Canaria (Spain, Canary Islands)"

_diversity, doi:10.3390/d13110551_

Round 1
Reviewer 1 Report
A very nicely written manuscript that resolves the identity of a weedy Persicaria on the Canary Islands. I really don't have much to say about the manuscript, the taxonomy is convincing and is a real strength but some of the ecological information needs critical reworking - too much detail.
Over all the English of the manuscript needs careful consideration, there is a tendency to use some words incorrectly, or inappropriately, such that the meaning of some sentences is hard to follow, or 'overly dramatic'. In places I have suggested some re-wordings to help with this however, as I am given to understand haste is needed (MDPI editor email) so I have refrained from an extensive rewording. This is not my role as a reviewer anyway.
I am not clear on the format either, e.g., are scientific names meant to be in italics or not for this journal? I started marking these up and then decided I might be missing something and stopped doing so.
The decision to make a new combination is justified but more could be made of this, especially in the abstract and keywords - otherwise that content will be lost in taxon searches.
I don't think a naturalisation that happened in the 1960s is in anyway 'recent' either.
Seriously though, these are minor points. Overall the manuscript is excellent, resolves the identity of a problematic weedy Persicaria, makes sensible decisions on its taxonomy, discusses its ecology and resolves its biostatus.
Well done.
I don't agree with reviewer anonymity either. I am happy for the authors to know my name.
Ciao
Peter J. de Lange
Author Response
A very nicely written manuscript that resolves the identity of a weedy Persicaria on the Canary Islands. I really don't have much to say about the manuscript, the taxonomy is convincing and is a real strength but some of the ecological information needs critical reworking - too much detail.
Correct, especially the information from La Palma was a bit too lengthy. I tried to be more concise but had to keep in mind that the connection with the Figures should not be lost.
Over all the English of the manuscript needs careful consideration, there is a tendency to use some words incorrectly, or inappropriately, such that the meaning of some sentences is hard to follow, or 'overly dramatic'. In places I have suggested some re-wordings to help with this however, as I am given to understand haste is needed (MDPI editor email) so I have refrained from an extensive rewording. This is not my role as a reviewer anyway.
I am not clear on the format either, e.g., are scientific names meant to be in italics or not for this journal? I started marking these up and then decided I might be missing something and stopped doing so.
Many of the scientific names were indeed not in italics. Corrected.
The decision to make a new combination is justified but more could be made of this, especially in the abstract and keywords - otherwise that content will be lost in taxon searches.
Correct. Abstract and keywords were modified accordingly.
I don't think a naturalisation that happened in the 1960s is in anyway 'recent' either.
I searched for ‘recent’ throughout the ms.: we nowhere wrote that the naturalization event is recent. The sudden increase of the species is a recent event, not its naturalization. I don’t think any changes are needed here.
Seriously though, these are minor points. Overall the manuscript is excellent, resolves the identity of a problematic weedy Persicaria, makes sensible decisions on its taxonomy, discusses its ecology and resolves its biostatus.
Well done.

Reviewer 2 Report
This is an excellent study of a cryptic invasive species in the Canary islands. The detailed morphological data, although not supported by molecular data (which are not conclusive) has allowed the authors to amend some misidentifications of Persicaria. As the species seems to be a fully invader (see my comments below), the present study may have practical applications. My only suggestions are the following:
- To make this ms. of more interest for the readers, I recommend to put into context the present case; the authors just mention in the Introduction that this is a case of a "cryptic invader" (lines 56-57). I would explain a little bit this concept and provide some examples in the Introduction. Also, I would also note the importance of uncover cryptic invaders within a wider strategy of IAS control. Indeed, I suggest the authors to focus the paper on this matter and present their results as a very nice case-study.
- Lines 375-388. The authors have used the categorization scheme of invasion stages of Richardson and colleagues, which is comprehensive and clear. However, the IUCN has recently proposed anouther standard ("EICAT"; https://www.iucn.org/theme/species/our-work/invasive-species/eicat), which is more based on the magnitude of impact. You could consider also using the latter.
Author Response
This is an excellent study of a cryptic invasive species in the Canary islands. The detailed morphological data, although not supported by molecular data (which are not conclusive) has allowed the authors to amend some misidentifications of Persicaria. As the species seems to be a fully invader (see my comments below), the present study may have practical applications. My only suggestions are the following:
To make this ms. of more interest for the readers, I recommend to put into context the present case; the authors just mention in the Introduction that this is a case of a "cryptic invader" (lines 56-57). I would explain a little bit this concept and provide some examples in the Introduction. Also, I would also note the importance of uncover cryptic invaders within a wider strategy of IAS control. Indeed, I suggest the authors to focus the paper on this matter and present their results as a very nice case-study.
Correct. In the Introduction we have added a paragraph that emphasizes the importance of cryptic invasions. Some further relevant references were also added.
Lines 375-388. The authors have used the categorization scheme of invasion stages of Richardson and colleagues, which is comprehensive and clear. However, the IUCN has recently proposed anouther standard ("EICAT"; https://www.iucn.org/theme/species/our-work/invasive-species/eicat), which is more based on the magnitude of impact. You could consider also using the latter.
We have checked the EICAT standard of the IUCN but feel that, at least in this particular case, the framework as proposed by Blackburn et al. is more appropriate. It was conceptualized by, among others, Pyšek and Richardson, two of the most important authorities when it comes to plant invasions in the past 2-3 decades.
